# Deep Self-expressive Learning

Chen Zhao[1], Chun-Guang Li[1], Wei He[1], and Chong You[2]

[1]School of Artificial Intelligence, Beijing University of Posts and Telecommunications, Beijing, China
[2]Department of EECS, University of California, Berkeley, CA, USA

*Self-expressive model* is a method for clustering data drawn from a union of low-dimensional *linear* subspaces. It gains a lot of popularity due to its 1) simplicity, based on the observation that each data point can be expressed as a linear combination of the other data points, 2) provable correctness under broad geometric and statistical conditions, and 3) many extensions for handling corrupted, imbalanced, and large-scale real data. This paper extends the self-expressive model to a Deep sELf-expressiVE model (DELVE) for handling more challenging case that the data lies in a union of *nonlinear* manifolds. DELVE is constructed from stacking multiple *self-expressive layers*, each of which maps each data point to a linear combination of the other data points, and can be trained via minimizing *self-expressive losses*. With such a design, the operator, architecture, and training of DELVE have the explicit interpretation of producing progressively linearized representations from the input data in nonlinear manifolds. Moreover, by leveraging existing understanding and techniques for self-expressive models, DELVE has a collection of benefits such as design choice by principles, robustness via specialized layers, and efficiency via specialized optimizers. We demonstrate on image datasets that DELVE can effectively perform data clustering, remove data corruptions, and handle large scale data.

## 1. Introduction

High-dimensional datasets in the forms of image, video, and text often have much fewer degrees of freedom than their measurement dimension, i.e., they are intrinsically low-dimensional [1]. While techniques such as the principal component analysis long existed for data with *linear* low-dimensional structures, practical data often lie in nonlinear manifolds. Moreover, complex data may contain multiple classes, hence are better modeled by multiple manifolds. Furthermore, it may be unknown a priori which data point lies in which manifold. Here we consider the problem of jointly learning a linearized representation and a clustering of data drawn from a union of nonlinear manifolds.

Many methods have been developed for learning linear representations for data in a nonlinear manifold, with extensions to handling multiple manifolds. Popular methods include Locally Linear Embedding [2], Laplacian Eigenmap [3], and many others, which are based on constructing a local neighborhood graph on the data and computing a linearized representation (i.e., low-dimensional embedding) that preserves desired properties of the graph. Although simple and intuitive, these methods require a large number of samples to reasonably estimate the local relationship, particularly when the samples are insufficient(i.e., not uniformly and densely sampled) or highly noisy, or the intrinsic dimension of the manifold is not low [4, 5].

Over the past decade, deep learning based methods, e.g., autoencoders [6–9], have become a powerful alternative with great empirical success. Based on performing a sequence of transforms (as opposed to a single-step transform in early methods), deep learning methods were argued to be more parameter-efficient for handling complex input-output relations [10]. Yet, popular deep architectures are constructed with generic layers, such as fully connected, convolutional, and ReLU, which are agnostic to the intrinsic structure(s) of the data. As a result, the network is usually deployed as a "black-box", lacking of understanding on the role of individual layers for learning with manifold(s).

First Conference on Parsimony and Learning (CPAL 2024).

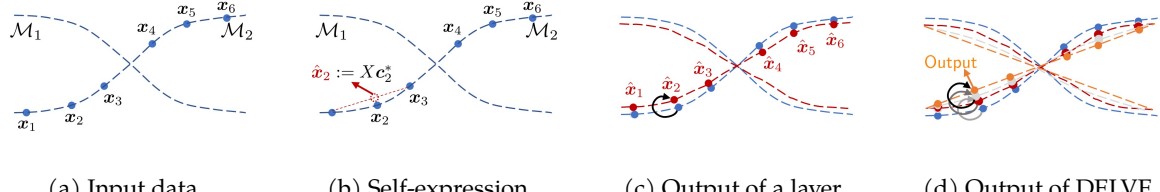

|  (a) Input data | (b) Self-expression | (c) Output of a layer | (d) Output of DELVE |

Figure 1: Conceptual illustration of DELVE. (a) Input data $X = [\boldsymbol{x}_1, \dots, \boldsymbol{x}_N]$ in a union of two manifolds $\mathcal{M}_1 \cup \mathcal{M}_2$ (points on $\mathcal{M}_1$ are not drawn for simplicity). (b) Inspired by the self-expressive model, we express each data point (with $\boldsymbol{x}_2$ as an example) as an approximate linear combination of other data points with coefficient vector $\boldsymbol{c}_2^*$ computed from problem (2). (c) With $\hat{\boldsymbol{x}}_2 := X\boldsymbol{c}_2^*$ computed, we illustrate the output of a self-expressive layer, which maps $\boldsymbol{x}_2$ to $\hat{\boldsymbol{x}}_2$ and likewise for all other data points. (d) Stacking self-expressive layers progressively on output of the previous layer produces linearized representations as output.

**A "Whitebox" Deep Manifold(s) Learning Model.** This paper presents a deep learning framework with a "white-box" design for learning manifolds, where each layer explicitly models the intrinsic low-dimensional manifold structures of the data. We draw inspiration from the *self-expressive* model [11], a popular tool for clustering data lying in a union of *linear* low-dimensional subspaces into their respective subspaces. Given a data matrix $X = [\boldsymbol{x}_1, \cdots, \boldsymbol{x}_N] \in \mathbb{R}^{D \times N}$, the self-expressive model exploits a simple observation that each point $\boldsymbol{x}_j$ can be expressed as a linear combination of other points in $X$ that are *from the same subspace as* $\boldsymbol{x}_j$, i.e., $\boldsymbol{x}_j = X\boldsymbol{c}_j$ where $\boldsymbol{c}_j \in \mathbb{R}^N$ with $c_{ij} \neq 0$ only if $\boldsymbol{x}_i, \boldsymbol{x}_j$ are in the same subspace. The coefficient vector $\boldsymbol{c}_j$ that satisfies such a property is referred to as being *subspace-preserving* and can be provably found by solving an optimization problem

$$\min_{\boldsymbol{c}_j} r(\boldsymbol{c}_j) \quad \text{s.t.} \quad \boldsymbol{x}_j = \sum_{i \neq j} c_{ij} \boldsymbol{x}_i, \tag{1}$$

with a proper choice of the regularizer $r(\cdot)$, e.g., the $\ell_1$ norm. Once the problem in (1) is solved, the coefficients $\{\boldsymbol{c}_j\}_{j=1}^N$ are used to construct a similarity graph for the data where the weight between $\boldsymbol{x}_i$ and $\boldsymbol{x}_j$ is set to $|c_{ij}| + |c_{ji}|$ by default, and spectral clustering [12] can be used to segment the data.

We now switch to the problem of interest of this paper, namely, the case where the columns of $X$ lie in a union of *nonlinear* manifolds. First, note that for data in nonlinear manifolds (see Fig. 1a), each data point $\boldsymbol{x}_j$ may not be expressed as an *exact* linear combination of other points from its own manifold. Nonetheless, a subspace-preserving $\boldsymbol{c}_j$ may still be found by solving the following problem with the constraint in (1) relaxed:

$$\boldsymbol{c}_j^* \doteq \arg\min_{\boldsymbol{c}_j} r(\boldsymbol{c}_j) + \gamma \cdot \kappa\Big(\boldsymbol{x}_j, \sum_{i \neq j} c_{ij} \boldsymbol{x}_i\Big), \tag{2}$$

where $\kappa(\cdot, \cdot)$ can be, e.g., the squared $\ell_2$ distance with $\gamma > 0$ being a trade-off parameter (see Fig. 1b). Put differently, while each manifold is globally nonlinear, at a local scale it is approximately linear. A key contribution of our work is to introduce a self-expressive layer from coefficients computed in (2), which performs a mapping $\Gamma : \boldsymbol{x}_j \mapsto \hat{\boldsymbol{x}}_j := X\boldsymbol{c}_j^*$ for each data point (see Fig. 1c). The intuition is that, if $\boldsymbol{c}_j^*$ is subspace-preserving, then the collection of output data points of the self-expressive layer is slightly more "linearized" than its input data points. Hence, by stacking multiple self-expressive layers, which gives rise to a deep architecture referred to as the *Deep sELf-expressiVE model* (*DELVE*), the underlying nonlinear manifolds are progressively transformed into linear subspaces, producing linear representations at the last layer (see Fig. 1d).

DELVE is a novel deep model with a "white-box" design. In particular, the operators, the architecture, and the training of DELVE are all explicitly designed to model the intrinsic low-dimensional structures. This allows us to understand the transformation performed by individual layer. In Fig. 2 we provide a visualization of the learned representation from DELVE on synthetic data on a single nonlinear manifold, and compare it with results from an autoencoder with $\tanh(\cdot)$ as the activation function. To enforce the autoencoder to learn low-dimensional embeddings, we set the dimension of encoder output to be 2. With DELVE, the output becomes increasingly linearized when more layers (i.e., larger $\ell$) are used, and with $\ell = 40$ the output lies closely on the unit sphere of a two-dimensional linear subspace. In contrast, the results with the autoencoder are less interpretable and desirable.

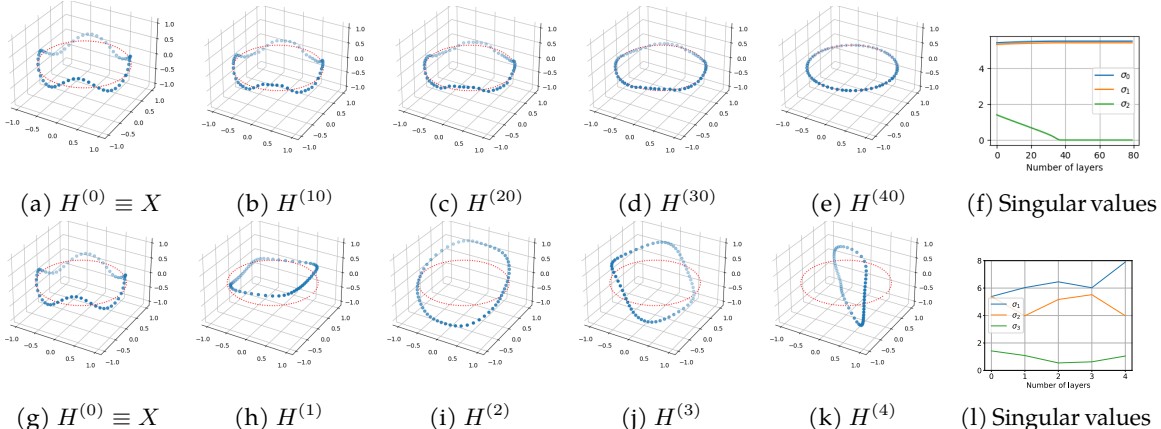

Figure 2: Visualization of learning a linearized representation for data lying in a nonlinear manifold of $\mathbb{R}^3$ by DELVE (top row) vs. autoencoder (bottom row). (a) - (e): Intermediate embeddings from DELVE at the $0, 10, 20, 30, 40$-th layers. (g) - (k): Intermediate embeddings at the $0, 1, 2, 3, 4$-th encoder of an autoencoder. (f) and (l): Singular values of the embeddings.

The benefits of the "white-box" design of DELVE compared to popular deep methods are as follows.

- Design Choices by Principles. Various design choices for DELVE can be made from principle. For example, the choice of $r(\cdot)$ in (2) controls a trade-off between obtaining subspace-preserving solutions and *connectedness* of points in each cluster, which can be examined by appropriate measures on the solution $c_j$, providing useful guidance for how to tune $r(\cdot)$ in practice.

- Robustness via Specialized Layers. Owning to the existence of many extensions of the self-expressive models to handling various types of data corruptions, DELVE may be made robust by adapting the design of the self-expressive layers accordingly. For example, by choosing $\kappa(\cdot, \cdot)$ in (2) to be $\ell_1$ norm, DELVE can be made robust to inputs corrupted with spurious noise.

- Efficiency via Specialized Optimizers. In DELVE, specialized optimizers may be developed to accelerate model training in optimizing (2). For example, active support methods [13, 14] may be used to gain efficiency by exploiting sparsity of the solution when $r(\cdot)$ is sparsity-promoting.

## 2. Related Work

**Manifold Learning and Denoising.** Classical methods for manifold learning are based on computing local neighborhood relationships using, e.g., linear coefficients that reconstruct each data point from its local neighbors [2, 15], local tangent space estimation [16], and heat kernels [3, 17]. Then, a linear representation can be obtained by spectral embedding (i.e., scaling method) [2, 3, 15, 17] or alignment techniques [16] that yield low-dimensional embeddings by preserving the local neighborhood relations. Similar ideas are used for manifold denoising using diffusion method [18] and robust Principal Component Analysis [19]. Our method is different from existing methods as we use a multi-layer self-expressive model to compute a sequence of neighborhood relationships with progressively linearized data and does not need spectral embedding or alignment technique.

**Self-Expressive Models and Subspace Clustering.** Prior work on the self-expressive model includes those that study the choice of regularization on the self-expressive coefficients [11, 14, 20–27], choice of data corruption penalty terms [22, 28–30], design of scalable algorithms [31–41], etc. Beside data clustering, the self-expressive model can be used for outlier detection [42, 43], filling in missing entries [44–47], exemplar selection [48, 49], concept discovery [50], etc. Many of the aforementioned techniques based on the self-expressive model have provable guarantees for correctness [42, 51–55] under mild conditions on the subspaces arrangement and data distribution. To handle nonlinear manifolds, self-expressive models may be extended by using the kernel trick [56], but it has the challenge of how to select an appropriate kernel. Recently, methods based on combining the self-expressive model with deep models to simultaneously learn feature representation and clustering of

data have become very popular [57–67]. However, there is lack of evidence that such an approach can produce the desired linear structures in the latent space [68].

**Whitebox Network Architectures.** In view of the lack of understanding of popular deep architectures there have been many alternative architectures with a "whitebox" design, e.g., [69–71], and reinterpretation of existing architectures, e.g., [72]. Among them the most relevant to our DELVE include those that learns low-dimensional linear representations [73–75]. However, [73, 74] requires labels to train and [75] only handles a single manifold. In contrast, our DELVE can handle multiple manifolds and does not require labels.

# 3. Deep Self-Expressive Model (DELVE)

**Problem Formulation.** Consider a data matrix $X = [\boldsymbol{x}_1, \ldots, \boldsymbol{x}_N] \in I\!\!R^{D \times N}$ with columns in a collection of $n$ unknown low-dimensional nonlinear manifolds $\{\mathcal{M}_1, \ldots, \mathcal{M}_n\}$. Assume that the data are unlabeled, i.e., the membership of the data points to each manifold is unknown. The goal is to simultaneously: 1) segment the columns of $X$ into their respective manifolds, and 2) learn a linearized representation for each manifold.[1]

## 3.1. Self-Expressive Layer

**Definition 3.1** (Self-expressive Layer). Given an input $H = [\boldsymbol{h}_1, \ldots, \boldsymbol{h}_N] \in \mathbb{R}^{D \times N}$, a self-expressive layer is a triplet $(C, \Gamma, \Omega)$, where $C = [\boldsymbol{c}_1, \ldots, \boldsymbol{c}_N] \in \mathbb{R}^{N \times N}$ is referred to as the *self-expressive coefficient matrix*, $\Gamma(H, C) \doteq HC$ is referred to as the *self-expressive map*, and

$$\Omega(H, C \mid r(\cdot), \gamma, \kappa(\cdot, \cdot)) = \sum_{j=1}^{N} \left( r(\boldsymbol{c}_j) + \gamma \cdot \kappa\Big(\boldsymbol{h}_j, \sum_{i \neq j} c_{ij} \boldsymbol{h}_i\Big) \right) \tag{3}$$

is referred to as the *self-expressive loss* where $r(\cdot) : \mathbb{R}^N \to \mathbb{R}$, $\gamma \in \mathbb{R}^+$, and $\kappa(\cdot, \cdot) : \mathbb{R}^D \times \mathbb{R}^D \to \mathbb{R}$ are design choices of the layer.

As in a neural network layer, the self-expressive layer defines an input-output mapping $H \mapsto \Gamma(H, C) := HC$ with $C$ being the matrix of the trainable parameters, hence can be used to construct a deep neural network. Unlike a neural network layer, the self-expressive layer itself comes with the self-expressive loss, which can be used for network training (explained in Section 3.2). The design idea of the self-expressive layer is explained in the introduction. Namely, if the columns of $H$ lie in a union of nonlinear manifolds each data point may be expressed *approximately* as a linear combination of a few other points from *its local neighborhood* in its own manifold. That is, there exist a matrix $C$ that satisfies $H \approx HC$, such that $C$ is *manifold-preserving*[2], i.e., the $i, j$-th entry of $C$ is nonzero only if $\boldsymbol{h}_i, \boldsymbol{h}_j$ are from the same manifold. Motivated by the case of linear subspaces, one may hope to recover such a representation as a solution $C^*$ to minimizing the self-expressive loss in (3), with $\gamma < \infty$ a finite real number, and $\kappa(\cdot, \cdot)$ as e.g., the Euclidean distance. Then, because $H \approx HC^*$, the self-expressive map produces the output of the layer $HC^*$ that deviates slightly from the layer input $H$. The intuition, as illustrated by an example in Fig. 1c, is that the output $HC^*$ linearizes the input $H$ slightly and thus reduces the curvature of the manifolds slightly.

Because the self-expressive layer has immediate interpretation as modeling the low-dimensionality of data, the particular design choices, such as $r(\cdot)$, $\gamma$, $\kappa(\cdot, \cdot)$, and algorithm for optimizing the self-expressive loss, can be made from principle. Moreover, the self-expressive layer can be customized for particular use cases, e.g., for handling corruptions in layer input. We explain the details below.

**Choice of $\gamma$.** As discussed above, a large value of $\gamma$ should be used when the manifolds are close to being linear, while smaller values of $\gamma$ should be used if the "degree" of nonlinearity is high.

---

[1]The problem is sometimes referred to as manifold clustering [76], or nonlinear subspace clustering [77].
[2]This term is adopted with its analogy to the notion of "subspace-preserving".

**Choice of $r(\cdot)$.** When the manifolds are linear, conditions under which the solution to minimizing (3) is subspace-preserving has been well-studied for many choices of $r(\cdot)$. The broadest conditions are established for $r(\cdot)$ being a sparse regularizer, such as the $\ell_1$ norm [42, 53, 78], making them the desired choice. However, sparse regularizers produce sparse solutions $C$, which is an undesirable property for the purpose of data clustering using the affinity graph $|C| + |C|^\top$ because data from the same manifold may not form a single connected component in this graph [79]. One of the successful strategies towards solving this problem is to set $r(\cdot)$ as the elastic net regularizer [14]:

$$r(\boldsymbol{c}_j) = \lambda \|\boldsymbol{c}_j\|_1 + \frac{1-\lambda}{2} \|\boldsymbol{c}_j\|_2^2, \tag{4}$$

where $\lambda \in [0, 1]$ is a trade-off parameter. Here, the introduction of the $\ell_2$ norm term has the effect of improving the density of the solution, motivated by earlier works [24, 80, 81]. Finally, while the aforementioned results have not been examined in the case of nonlinear manifolds, we may still use them as the guiding principle for choosing $r(\cdot)$ in self-expressive layers.

**Choice of $\kappa(\cdot, \cdot)$.** Different choices of $\kappa(\cdot, \cdot)$ are used to handle different types of corruptions in data. For example, the $\ell_2$ norm is used to handle dense Gaussian noise [82], while the $\ell_1$ norm is used to handle sparse corruptions [83]. In the self-expressive layer and when the underlying manifolds in the input $H$ are nonlinear, $\kappa(\cdot, \cdot)$ may be used to handle corruptions in $H$ in the same way as in linear subspaces; However, it also needs to model the deviation of the underlying manifold from linearity, i.e., the fact that a point on a manifold cannot be expressed as an exact linear combination of its neighboring points. Motivated by the form of the elastic net regularizer in (4), in this paper, we will use the penalty term

$$\kappa(\boldsymbol{h}, \hat{\boldsymbol{h}}) = \eta \|\boldsymbol{h} - \hat{\boldsymbol{h}}\|_1 + \frac{1-\eta}{2} \|\boldsymbol{h} - \hat{\boldsymbol{h}}\|_2^2, \tag{5}$$

where the first term may be used if the input is expected to have sparse corruptions, and the second term may be used if the input is expected to have dense Gaussian noise, $\eta > 0$ is a tuning parameter.

**Choice of optimization algorithm.** Many optimization algorithms were developed for minimizing the self-expressive loss in (3) depending on the choice of $r(\cdot)$ and $\kappa(\cdot, \cdot)$. For a sparse regularizer $r(\cdot)$, e.g., the elastic net regularizer in (4) with a large $\lambda$ close to 1, there are active support based algorithms [14, 84, 85] that leverage such sparsity for performing fast computation, and can handle relatively large scale datasets with $100,000$ data points. In this paper, we use the solver in [14]. We also mention that there also exists many approximate algorithms based on subsampling [31], sketching [35], divide-and-conquer [86] and beyond, which have $O(N)$ complexity and can handle datasets with a million points. These algorithms may be used to further scaling up our DELVE.

## 3.2. DELVE: Architecture and Training

We now introduce DELVE which is a deep learning model built upon self-expressive layers. Denote $H^{(0)} \doteq X$ the model input. Then, the architecture of DELVE is obtained from alternatively applying self-expressive layers and normalization layers to the input. That is,

$$\widehat{H}^{(\ell+1)} = H^{(\ell)} C^{(\ell+1)}, \qquad H^{(\ell+1)} = \Pi(\widehat{H}^{(\ell+1)}), \tag{6}$$

for $\ell = 0, \ldots, L-1$. In above, $\{C^{(\ell)}\}_{\ell=1}^L$ are the self-expressive coefficient matrices of $L$ self-expressive layers, which are trainable parameters. $\Pi(\cdot)$ is a mapping that normalizes each column of its input by its $\ell_2$ norm. Without it we empirically find that the representations tend to collapse to the origin.

Because each self-expressive layer has a loss function, DELVE can be trained in an unsupervised way. Specifically, given any $\{(r^{(\ell)}(\cdot), \gamma^{(\ell)}, \kappa^{(\ell)}(\cdot, \cdot))\}_{\ell=1}^L$ associated with the $L$ self-expressive layers, DELVE may be trained by solving the following optimization problem:

$$\min_{C^{(1)}, \ldots, C^{(L)}} \sum_{\ell=1}^L \Omega^{(\ell)}\Big(H^{(\ell-1)}, C^{(\ell)}\Big) \doteq \sum_{\ell=1}^L \Omega\Big(H^{(\ell-1)}, C^{(\ell)} \mid r^{(\ell)}(\cdot), \gamma^{(\ell)}, \kappa^{(\ell)}(\cdot, \cdot)\Big). \tag{7}$$

We use a layer-wise training algorithm for optimizing (7). That is, the weights from shallower layers to deeper layers, i.e., $C^{(1)}, C^{(2)}, \ldots, C^{(L)}$ are trained sequentially. Moreover, when optimiz-

---

**Algorithm 1** Layer-wise Training Procedure for DELVE.

---

**Input:** Data $X = [\boldsymbol{x}_1, \cdots, \boldsymbol{x}_N]$, number of layers $L$, $\{(r^{(\ell)}(\cdot), \gamma^{(\ell)}, \kappa^{(\ell)}(\cdot, \cdot))\}_{\ell=1}^L$.

1: Initialize $\ell = 1$, $H^{(\ell-1)} = X$;
2: **while** $\ell \leq L$ **do**
3:     Compute $C^{(\ell)*} = \arg\min_{C^{(\ell)}} \Omega(H^{(\ell-1)}, C^{(\ell)} \mid r^{(\ell)}(\cdot), \gamma^{(\ell)}, \kappa^{(\ell)}(\cdot, \cdot))$;
4:     Set $\widehat{H}^{(\ell)} = H^{(\ell-1)} C^{(\ell)*}$;
5:     Set $H^{(\ell)} = \Pi(\widehat{H}^{(\ell)})$ from dividing each column of $\widehat{H}^{(\ell)}$ by its $\ell_2$ norm;
6:     $\ell \leftarrow \ell + 1$;
7: **end while**
**Output:** Learned representation $H^{(L)}$ and learned network parameters $\{C^{(\ell)}\}_{\ell=1}^L$.

---

ing $C^{(\ell)}$, while all loss terms in deeper layers, i.e., $\{\Omega^{(\ell')}(H^{(\ell'-1)}, C^{(\ell')}\}_{\ell' \geq \ell}$ are a function of $C^{(\ell)}$ (because $H^{(\ell'-1)}$ depends on $C^{(\ell)}$ for all $\ell' > \ell$), we only minimize the loss term at the $\ell$-th layer, i.e., $\Omega^{(\ell)}(H^{(\ell-1)}, C^{(\ell)})$. We summarize the layer-wise training procedure for DELVE in Algorithm 1. Finally, we note that (7) may be solved (or fine-tuned after the layer-wise training) by other algorithms, such as gradient descent on gradients calculated via back-propagation. We defer a study of such alternative training techniques to future work.

Once DELVE is trained, the output from the last layer $H^{(L)}$ is the learned representation for the input data $X$. In addition, the learned parameters $C^{(L)}$ in the last layer can be used to construct a similarity graph for the data with weight matrix $|C^{(L)}| + |C^{(L)}|^\top$. We can then apply spectral clustering to the similarity graph to obtain the membership of columns of $X$ to the manifolds.

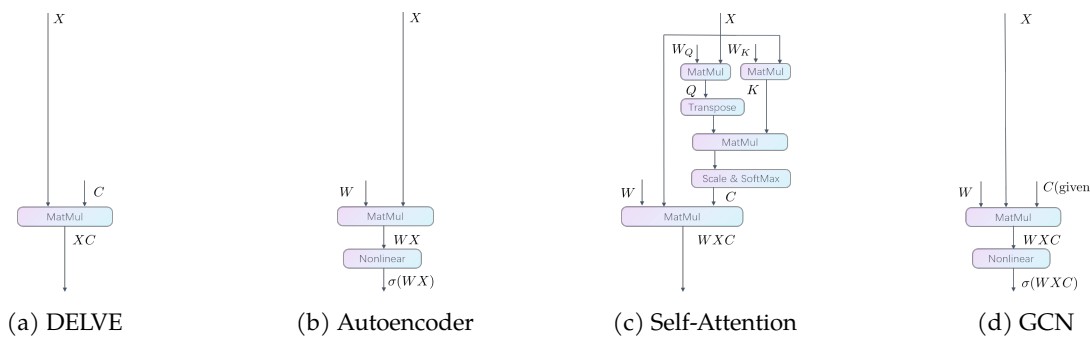

(a) DELVE          (b) Autoencoder          (c) Self-Attention          (d) GCN

Figure 3: Comparison of building block of DELVE with those of related deep learning architectures.

## 3.3. Related Architectures

Comparing with popular deep models for manifold learning, such as autoencoders, DELVE is fundamentally different. Specifically, given a layer input $X$, each layer of DELVE computes a linear combination of all data points, i.e., $XC$, see Fig. 3a. In contrast, each layer of an autoencoder performs a linear mapping on each data point, i.e., $WX$, see Fig. 3b. From this perspective, DELVE is more similar to self-attention models and graph convolutional networks as explained next.

**Connection to Self-Attention Models.** Self-attention is a broadly used building block for Transformers [87], Graph Attention Networks [88], and so on. Given a sequence of input vectors $X = [\boldsymbol{x}_1, \ldots, \boldsymbol{x}_N]$, a self-attention model computes a sequence of output vectors where each vector is a linear combination of the input vectors, i.e., $XC$, with $C$ being an $N \times N$ matrix. Such a computation is the same as the self-expressive map performed by self-expressive layers, see Definition 3.1.[3] The main difference lies in how the matrix $C$ is computed. In self-attention models, $C$ is computed by the inner product of the embeddings from the key and query followed by a softmax layer. As illustrated in Fig. 3c, the key $K$ and query $Q$ are computed from parameterized functions of $X$ in which the

---

[3]Broadly, the connection of self-attention and self-expressive models has been noted in [41, 89].

| Methods | BP | DA | EYaleB | | | COIL100 | | | MNIST | | |
|---|---|---|---|---|---|---|---|---|---|---|---|
| | | | ACC | NMI | ARI | ACC | NMI | ARI | ACC | NMI | ARI |
| $k$-means | | | 9.47 | 13.02 | 1.31 | 47.72 | 74.96 | 42.47 | 55.29 | 49.17 | 37.42 |
| SC [12] | | | 42.13 | 56.41 | 13.71 | 53.28 | 81.24 | 24.67 | 67.23 | 79.81 | 62.54 |
| SSC [11] | | | 67.98 | 74.85 | 20.54 | 67.84 | 73.74 | 19.32 | 79.47 | 78.33 | 69.49 |
| LSR [24] | | | 70.51 | 75.40 | 51.37 | 46.74 | 72.24 | 38.92 | - | - | - |
| LRSC [23] | | | 69.40 | 74.56 | 50.95 | 49.39 | 74.48 | 43.01 | - | - | - |
| EnSC [14] | | | 68.18 | 74.46 | 19.45 | 66.11 | 89.31 | 63.22 | _83.27_ | _82.67_ | _76.12_ |
| SSC-OMP [13] | | | 79.91 | 83.78 | 53.04 | 31.46 | 58.20 | 15.25 | 47.41 | 48.77 | 34.94 |
| S³COMP [40] | | | _84.25_ | _85.27_ | _68.05_ | 75.19 | 93.06 | 72.29 | 63.77 | 65.58 | 52.33 |
| LLMC [94] | | | 55.68 | 64.88 | 28.54 | 74.39 | 92.29 | 70.14 | - | - | - |
| KSSC [56] | | | 72.25 | - | - | 62.72 | - | - | - | - | - |
| SMCE [95] | | | 56.20 | 60.72 | 29.83 | _80.01_ | **94.32** | **76.80** | - | - | - |
| AE+SSC [58] | ✓ | | 74.67 | - | - | 56.07 | - | - | - | - | - |
| AE+LSR [68] | ✓ | | 71.96 | - | - | 44.84 | - | - | - | - | - |
| DSCNet-$\ell_2$ [58, 68][4] | ✓ | | 59.09 | - | - | 45.67 | - | - | - | - | - |
| DSC-$\ell_1$ [63] | ✓ | | - | - | - | 69.63 | 87.51 | 58.35 | - | - | - |
| DSC-$\ell_2$ [63] | ✓ | | - | - | - | 69.03 | 86.75 | 60.01 | - | - | - |
| NMCE [96] | ✓ | ✓ | - | - | - | **88.47** | - | - | - | - | - |
| **DELVE** (ours) | | | **89.76** ±1.05 | **90.12** ±0.19 | **73.64** ±1.47 | 78.96 ±0.46 | _93.86_ ±0.04 | _75.81_ ±0.39 | **96.38** ±0.05 | **90.95** ±0.14 | **92.19** ±0.09 |

Table 2: Clustering performance (%) on EYaleB, COIL100, and MNIST. **Best** and second best results are highlighted. BP: method requires back-propagation. DA: method requires data augmentation.

parameters $W_Q, W_K$ can be learned from training data. In contrast, in DELVE, $C$ itself is the trainable parameter that can be learned from data.

**Connection to Graph Convolution Networks (GCNs).** GCN [90] is a popular model for learning from data associated with graphs. Consider a graph $\mathcal{G}(\mathcal{V}, \mathcal{E}, C)$ where $\mathcal{V}$ is the set of vertices, $\mathcal{E}$ is the set of edges and $C$ is an adjacency matrix for the vertices. Each vertex $v_j$ in $\mathcal{V}$ is associated with an embedding vector $\boldsymbol{x}_j$. One of the most important operations in GCN is to aggregate information for each vertex $v_j$ from its neighbors on the graph. This is achieved by updating the embedding vector associated with vertex $v_j$ by the summation of the embedding vectors of its neighbors. Collecting the embeddings of all vertices as matrix $X = [\boldsymbol{x}_1, \ldots, \boldsymbol{x}_N]$, such an aggregation operation can be represented as $XC$, which is the same as the self-expressive map in self-expressive layers (see Fig. 3d). Hence, the self-expressive layer can be interpreted as a generalization of the information aggregation layer, where the adjacency matrix $C$ is learned from data, rather than given and fixed.

## 4. Experiments

To verify the effectiveness of DELVE, we conduct experiments on datasets COIL-100 [91], Extended Yale B (EYaleB) [92], and MNIST [93], for which the key information is summarized in Table 1. Details on datasets, DELVE implementation, and comparing methods are provided in Appendix C. Results on additional datasets are provided in Appendix B.3.

### 4.1. Evaluating DELVE for Data Clustering

We report the clustering performance measured by clustering accuracy (ACC), normalized mutual information (NMI), and adjusted rand index (ARI) (see e.g. [97] for their definitions) in

| Datasets | # Total samples | # Classes | Image size | Feature dimension |
|---|---|---|---|---|
| COIL100 | 7,200 | 100 | $32 \times 32$ | 1,024 |
| EYaleB | 2,432 | 38 | $48 \times 42$ | 2,016 |
| MNIST | 70,000 | 10 | $28 \times 28$ | 784 |

Table 1: Summary of dataset information.

Table 2. It can be seen that DELVE is the best performing method on EYaleB and MNIST. On COIL100, DELVE is the third best performing method following SMCE [95] and NMCE [96]. Here, we mention that SMCE is a method based on the self-expressive model with an additional locality constraint. Motivated by SMCE, we also trained a version of DELVE, referred to as DELVE-SMCE, with such a constraint added to the self-expressive loss in (3). The ACC, NMI, and ARI of DELVE-SMCE on COIL100 are given by $84.64\%$, $94.38\%$, and $81.94\%$, respectively, which are considerably higher than those of SMCE. In a nutshell, with a proper choice of self-expressive loss, DELVE is the best performing method on COIL100 except for NMCE [96]. The better performance of NMCE relies critically on its use of data augmentation, for which our DELVE does not use.

**DELVE is Scalable.** We highlight an advantage of DELVE over many recent deep subspace clustering methods, namely, DELVE can effectively handle large-scale datasets. As shown in Table 2, the results for DSCNet and DSC are not listed for MNIST because they cannot handle 70,000 data points. The reason is that these methods require full batch training via back-propagation, where all data points and all their intermediate feature maps need to be loaded into the memory.

| Methods | ACC | NMI | ARI |
|---|---|---|---|
| $k$-SCN-G [98] | 82.22 | 73.97 | 71.10 |
| $k$-SCN-S [98] | 87.14 | 78.15 | 75.81 |
| PSSC [66] | 89.00 | 79.00 | - |
| EDESC [99] | 91.30 | 86.20 | - |
| **DELVE** (ours) | **96.38** | **90.95** | **92.19** |

Table 3: Clustering accuracy (%) on MNIST compared to scalable methods.

In contrast, with the layer-wise training strategy for DELVE, the intermediate feature maps of already trained layers can be removed from the memory when training subsequent layers. Finally, while there are recently developed scalable deep subspace clustering methods, such as $k$-SCN-G and $k$-SCN-S [98], PSSC [66], and EDESC [99], their performance lags far behind our method (see Table 3).

**Effect of Model Depth.** We evaluate how clustering accuracy changes across layers. We take 20 classes from COIL-100 (a.k.a., COIL-20) and plot the results under varying choices of parameter $\gamma$ for the self-expressive loss (3) in Fig. 4a. In a wide range of $\gamma$, the clustering accuracy increases rapidly in the first few layers which shows the effectiveness of DELVE. Meanwhile, the clustering accuracy starts to drop after a certain layer, showing an over-fitting issue. We leave a study on the cause of overfitting to future work. In Fig. 4b, we report self-expressive residual (SER), defined as $\frac{\|H^{(\ell)}-H^{(\ell-1)}\|_F^2}{\|H^{(\ell-1)}\|_F^2}$, which measures

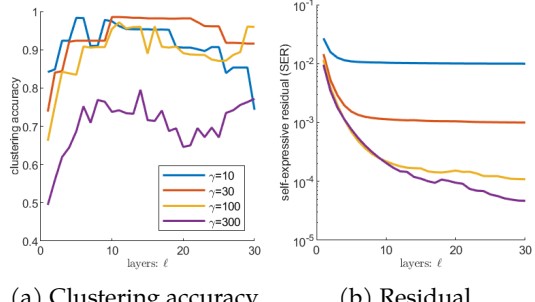

(a) Clustering accuracy  (b) Residual

Figure 4: Evaluation of DELVE on COIL20 across layers under varying choices of $\gamma$.

the relative changes from the output to the input of layer $\ell$. We see that SER progressively decreases as the number of iteration increases. This shows that the effect of adding more layers in DELVE is diminishing. Such a behavior is qualitatively similar to that observed in residual networks [100].

## 4.2. Understanding and Customizing DELVE

As discussed in Section 3.1, DELVE can be customized to accommodate different use cases with proper choices of the regularizer $r(\cdot)$, the penalty term $\kappa(\cdot, \cdot)$, and the optimization algorithm.

**Understanding the Choice of Regularization Term** $r(\cdot)$**.** As discussed in Section 3.1, the property of the learned weight matrices $\{C^{(\ell)}\}$ relies on the choice of the regularization $r(\cdot)$. In particular, $\{C^{(\ell)}\}$ should neither be too sparse, as this may hurt within-class connectivity, nor be too dense, as this may lead to violation of the subspace-preserving (i.e., manifold-preserving) property. With $r(\cdot)$ being the elastic net regularizer in (4), such a trade-off is controlled by the hyper-parameter $\lambda$. In the following, we explain how the effect of $\lambda$ on DELVE can be observed and measured explicitly for gaining insights into how DELVE performs. Specifically, we consider the following three metrics.

- Subspace-representation error (SRE) [83]: The proportion of magnitude of entries in $C^{(\ell)}$ that comes from the wrong class, i.e., $\frac{1}{N} \sum_j \left( \sum_i w_{ij} |c_{ij}^{(\ell)}| / \|\boldsymbol{c}_j^{(\ell)}\|_1 \right)$ where $w_{ij}$ is 1 if $i, j$ are in different classes and 0 otherwise. If $C^{(\ell)}$ is subspace-preserving, then its SRE is 0.

- Connectivity [13]: We compute the algebraic connectivity [101] of graph Laplacian defined from $C^{(\ell)}$ for each class and report the minimum over all classes. Hence, this measure is zero if and only if there exist a class that the similarity graph associated with $C^{(\ell)}$ is not connected.

- Sparsity: It measures the average number of nonzero entries per columns in $C^{(\ell)}$.

---

[4]The DSCNet [58] uses a very complex postprocessing step that is highly tuned on EYaleB. However, [68] shows that adding the same postprocessing significantly improves other self-expressive methods. In order to have a fair comparison with all methods, we follow [68, 77] to report results without such a postprocessing.

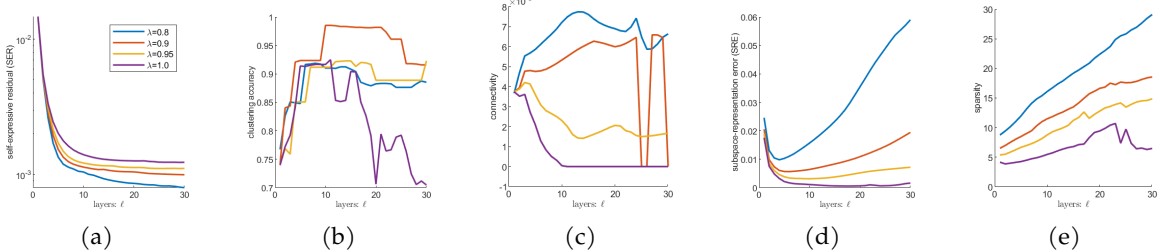

(a)                (b)                (c)                (d)                (e)

Figure 5: Effect of $\lambda$ in the regularizer $r(\cdot)$ on DELVE using COIL20.

We use COIL20 for the evaluation and report the three metrics listed above, as well as ACC and SER, in Fig. 5. From Fig. 5c, using $\lambda = 1.0$ leads to a zero connectivity after 10 iterations, which explains its inferior clustering performance demonstrated in Fig. 5b. By reducing $\lambda$, the sparsity increases as shown in Fig. 5e and we obtain a better connected solution $C^{(\ell)}$ which leads to a better and more stable clustering performance. Yet, setting $\lambda$ to be too small increases SRE as shown in Fig. 5d hence worse clustering performance. This example shows that the effect of $\lambda$ for DELVE can be examined in a principled way by measuring the properties of learned weights.[5]

**Robustness by Choice of Penalty Term** $\kappa(\cdot, \cdot)$. We demonstrate that DELVE can be customized to handle sparse corruptions by a proper choice of the penalty term $\kappa(\cdot, \cdot)$. We randomly pick 30% of all images in EYaleB, and add sparse (i.e., pepper-and-salt) noise to $p \in \{0, 20, 40, 60, 80, 100\}$ percent of the pixels in each of the selected images. To model sparse corruptions, we use $\kappa(\boldsymbol{x}, \hat{\boldsymbol{x}}) = \|\boldsymbol{x} - \hat{\boldsymbol{x}}\|_1$ in each layer of DELVE, and report the clustering accuracy in Fig. 6. We can observe that our DELVE is robust to the corruption with a very small performance drop as long as the percentage of pixel corruptions $p\%$ is less than 70%. This behavior is in sharp contrast to DSCNet [58, 68], a deep learning based approach where the performance drops rapidly even when $p\%$ is small. We provide a visualization of the learned representations at the output of DELVE under varying corruption level in Fig. B.1 (see appendix).

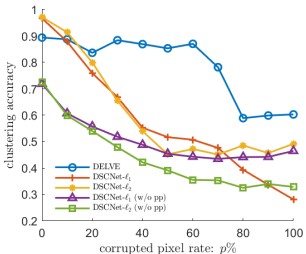

Figure 6: Robustness Evaluation of DELVE to sparse corruptions on EYaleB.

# 5. Conclusion and Discussion

This paper presented the Deep sELf-expressiVE model (DELVE) for jointly learning a linearized representation and performing clustering of data drawn from a union of low-dimensional nonlinear manifolds. Different from the conventional self-expressive model that operates on the original data and is inherently shallow, DELVE is built upon a multi-layer architecture where each layer linearizes the data by performing self-expression and the entire model operates on progressively linearized data. Such a construction is "'whitebox' in nature and opens up the possibility of leveraging the rich results developed for the self-expressive model, including both theory and practical algorithms, for the development of next generation deep models that are more amenable to theoretical study and principled design. While this paper cannot fully explore all such potentials, our experiments already demonstrate the promising features in terms of explainability and robustness compared to existing approaches. Moving forward, we believe that many other techniques in self-expressive models such those developed for handling outliers, missing entries, imbalanced data, large-scale data can be leveraged to further improve the capability of DELVE.

**Acknowledgment** C. Zhao, C.-G. Li, and W. He are supported by the National Natural Science Foundation of China under Grant 61876022. C. You is currently at Google Research, New York City, and contributed to this work while he was at UC Berkeley and supported by the Tsinghua-Berkeley Shenzhen Institute Research Fund. We acknowledge helpful comments from Benjamin D. Haeffele from Johns Hopkins University.

---

[5]This is in contrast to deep learning where it is hard to understand the effect of hyper-parameters.

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

# Appendices

The appendices are organized as follows. In Section A we provide implementation details for the results in Figure 2, and additional results for the case where the data are corrupted by noise. In Section B we provide additional experiments on real datasets. In Section C we provide description of datasets and implementation details for experiments on real datasets. In Section D we provide the implications and limitations of our proposed DELVE.

## A. Additional Materials for Experiments on Synthetic Data

### A.1. Implementation Details

We provide details for results reported in Figure 2.

The data are generated by sampling 60 data points from
$$(x, y, z) = (\cos(\theta)\cos(\phi), \cos(\theta)\sin(\phi), \sin(\theta)), \tag{A.1}$$
with $\theta = \frac{\pi}{12}\sin(4\phi)$ and $\phi$ taken uniformly from $[0, 2\pi]$. Our DELVE is trained with the elastic net regularizer in Eq. (4) and a penalty term $\kappa(\boldsymbol{x}, \hat{\boldsymbol{x}}) = \frac{1}{2}\|\boldsymbol{x} - \hat{\boldsymbol{x}}\|_2^2$ in each self-expressive layer, where we fix $\lambda = 0.8$ and $\gamma = 100$.

### A.2. Results on Corrupted Data

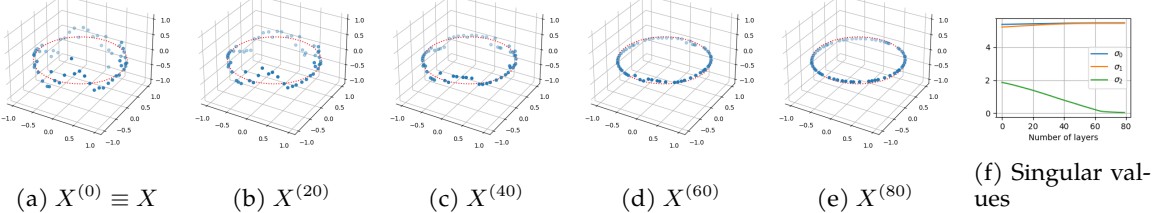

(a) $X^{(0)} \equiv X$     (b) $X^{(20)}$     (c) $X^{(40)}$     (d) $X^{(60)}$     (e) $X^{(80)}$     (f) Singular values

Figure A.1: Visualization of learning a clean representation for corrupted data that lies approximately in a nonlinear manifold in $\mathbb{R}^3$. (a) 60 data points in a nonlinear manifold corrupted by noise. (b)-(e): Output from DELVE at the $20, 40, 60, 80$-th layers. (f): Singular values.

We further test the ability of DELVE in handling cases where the data are corrupted by noises hence do not lie exactly on the nonlinear manifold. Specifically, we add i.i.d. Gaussian noises with zero mean and standard deviation $0.2$ to $\theta$ when generating data points with Eq. (A.1) (see Fig. A.1 (a)). The outputs of DELVE at different layers are displayed in Fig. A.1 (b)-(e). We can observe that the noises are progressively filtered out and the outputs are progressively linearized, though it took more iterations to achieve so than with clean data. The singular values of the output data at different layers, as shown in Fig. A.1 (f), confirm that DELVE produces linearized representations.

It is worth to note that we did not compare with a 40-layer autoencoder because: a) there is difficulty to train it, and b) increasing the number of layers for autoencoder does not qualitatively change the results. On the other hand, we are showing the result for DELVE at the 40th layer rather than the 4th layer because DELVE cannot produce linear representations with only 4 layers. This does not mean that our comparison of a 40-layer DELVE to a 4-layer autoencoder is unfair.

## B. Additional Experimental Results on Real Data

### B.1. Experiments with Other Choices of $r(\cdot)$

In Section 4, all our results for DELVE are obtained with $r(\cdot)$ being the elastic net regularizer. Here we show that other choices of regularizers can also be adopted. In particular, we conduct experiments with the following choices of $r(\cdot)$.

| Methods | Extended Yale B | | | COIL100 | | | MNIST | | |
|---|---|---|---|---|---|---|---|---|---|
| | ACC | NMI | ARI | ACC | NMI | ARI | ACC | NMI | ARI |
| SSC | 67.98 | 74.85 | 20.54 | 67.84 | 73.74 | 19.32 | 79.47 | **78.33** | 69.49 |
| **DELVE**-SSC (ours) | **88.53** | **89.03** | **66.11** | **78.61** | **93.54** | **71.99** | **81.58** | 72.55 | **80.37** |
| SMCE | 56.20 | 60.72 | 29.83 | 80.01 | 94.32 | 76.80 | - | - | - |
| **DELVE**-SMCE (ours) | **67.26** | **71.93** | **32.26** | **84.64** | **94.38** | **81.94** | - | - | - |

Table 4: Clustering performance of DELVE with varying choices of regularizer $r(\cdot)$. "-" means that the method cannot handle large scale data.

- The $\ell_1$-norm. This is used in SSC [11].

- The $\ell_1$-norm with a locality constraint, i.e., $c_{ij} = 0$ if $h_j$ is not a $k$-nearest neighbor of $h_i$, where $k$ is a hyper-parameter. This is used in SMCE [95].

We refer to the methods above to be DELVE-{SSC, SMCE}, respectively. To distinguish from these three methods, we will refer to the method of DELVE in Table 2 as DELVE-EnSC when there is risk of confusion.

Experimental results are shown in Table 4.

Recall that from the architecture of DELVE, the methods {SSC, SMCE} are special cases of DELVE-{SSC, SMCE } with only one layer. Hence, as DELVE-{SSC, SMCE} improves upon {SSC, SMCE}, respectively, in all three metrics of ACC, NMI, and ARI, the result demonstrates the benefit of having multiple layers, i.e., the effectiveness of the architecture of DELVE. In addition, comparing the results reported in Table 4 with those reported in Table 2, we can see that DELVE-SMCE is the best performing method on COIL100 among all comparing methods.

## B.2. Visualization of Corruption Removal on EYaleB

To have a clear perception of the robustness of our DELVE, we visualize the learned representations at the output of DELVE by reshaping each representation vector into an image and display in Fig. B.1 a subset of such learned representations under varying corruption level. It can be seen that DELVE provides a recovery of the clean image as long as the percentage of corruption is no more than 60%.

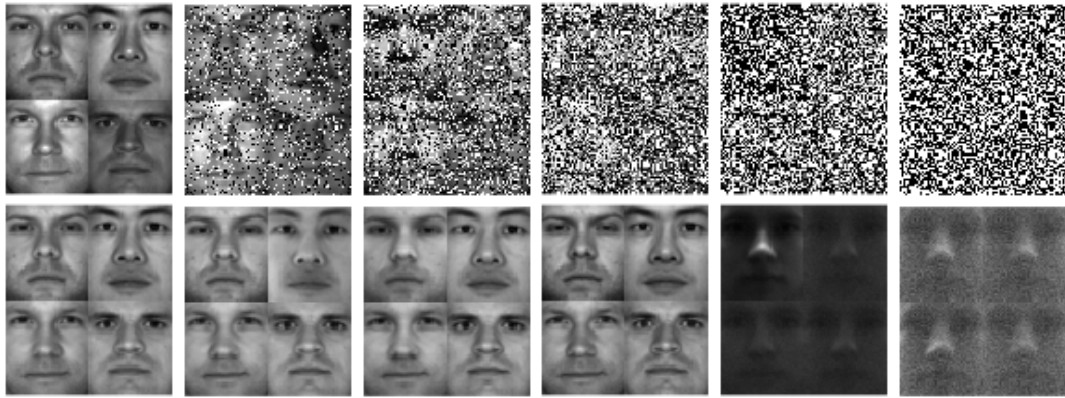

Figure B.1: Visualization for Corruption Removal. Top: Selected 4 samples from 4 different classes corrupted by $p \in \{0\%, 20\%, 40\%, 60\%, 80\%, 100\%\}$ pixels. Bottom: Corresponding output at the last layer of DELVE.

## B.3. Experiments on CIFAR10 and Fashion-MNIST

To further evaluate the performance of DELVE, we conduct a set of experiments on two additional datasets, namely CIFAR-10 and Fashion-MNIST datasets.

**CIFAR-10** contains 60,000 color images in 10 classes, where each image is of size $32 \times 32$. In experiments, we use the feature representation extracted by MCR$^2$ [97], which aims to learn feature representation lying in a union-of-subspace with a self-supervised learning strategy. **Fashion-MNIST** contains 70,000 grey-scale images of various types of fashion products. Fashion products (e.g., coat, trouser, shirt, dress, bag, etc.) with different styles correspond to 10 categories. We compute a feature vector of dimension 3,472 using the scattering convolution network [102], which extracts translation-invariant and deformation-stable features, and then we reduce the dimension to $500$ via PCA. All feature vectors are normalized to have unit $\ell_2$ norm.

The results are reported in Table 5, where we compare DELVE with EnSC and a few deep clustering methods and deep subspace clustering methods, including DAC [103], $k$-SCN-S [98], NCSC [62], ClusterGAN [104]. We can observe that DELVE-EnSC yields notable improvement over EnSC, and outperforms other competing methods.

| Methods | CIFAR-10 | | | FashionMNIST | | |
|---|---|---|---|---|---|---|
| | ACC | NMI | ARI | ACC | NMI | ARI |
| DAC [103] | 52.20 | 39.60 | 30.60 | - | - | - |
| ClusterGAN [104] | - | - | - | 66.20 | 64.50 | - |
| DCCM [105] | 62.30 | 49.60 | 40.80 | - | - | - |
| $k$-SCN-S[98] | - | - | - | 60.02 | 62.30 | - |
| NCSC[62] | - | - | - | 72.10 | 68.60 | 59.20 |
| EnSC | 66.43 | 63.01 | 44.07 | 72.04 | 68.40 | 56.82 |
| **DELVE**-EnSC (ours) | **68.70** | **63.57** | **47.13** | **74.01** | **71.14** | **62.09** |

Table 5: Comparison of clustering performance on CIFAR-10 and Fashion-MNIST. The results of comparing methods (other than EnSC) are taken from their respective papers.

# C. Implementation Details for Experiments on Real Data

## C.1. Description of Datasets

**EYaleB** is a dataset that contains frontal face images of $38$ individuals where each individual is taken under $64$ different illumination conditions. Hence, the images associated with each face can be modeled as a 9 dimensional linear subspace plus sparse corruption, where the linearity comes from the characterization of reflection of Lambertian object and the sparse corruption is due to deviations from Lambertian property caused by e.g., shadows and specularities. In our experiments, we resize each image from the original size of $192 \times 168$ pixels into $48 \times 42$ pixels and concatenate the gray pixels as a 2016-dimensional vector.

**COIL-100** is an image dataset of 100 different objects. For each object, 72 images were taken by rotating the object on a turnable through 360 degrees with an interval of 5 degrees. Hence, each class lies closely to a one-dimensional nonlinear manifold. We convert the images to gray-scale, resize each image to the size $32 \times 32$, and concatenate the pixel values in each image as a 1024-dimensional vector.

**MNIST** contains 70,000 gray-scale images of handwritten digits "0" to "9" of size $28 \times 28$. In our experiments, we concatenate the gray pixels in each image as a 784-dimensional vector, without other feature extraction step.

## C.2. Details for Comparing Methods

We evaluate DELVE by its performance on clustering the datasets, and compare it with the following 5 categories of methods. 1) Standard clustering methods, including $k$-means [106] and spectral clustering (SC) [12]. 2) Subspace clustering methods, including SSC [11], LSR [24], LRSC [107], EnSC [14], SSCOMP [13], and S$^3$COMP [40]. 3) (Shallow) manifold clustering methods, including KSSC [56], LLMC [94], and SMCE [95]. 4) Autoencoder + subspace clustering methods, including AE+SSC and AE+LSR, where a convolutional autoencoder extracts the features on which subspace

clustering methods are applied. 5) Deep subspace clustering methods that jointly trains a deep network for feature extraction and clusters the features, including DSCNet-$\ell_2$ [58], DSC-$\ell_1$ / -$\ell_2$ [63], and NMCE [96].

We provide implementation details for these methods in Table 2.

For Spectral Clustering (SC) [12], we use the similarity matrix computed from an adaptive Gaussian kernel:

$$a_{ij} = \exp\left(-\frac{\|\boldsymbol{x}_i - \boldsymbol{x}_j\|_2^2}{\sigma_i^{(k)}\sigma_j^{(k)}}\right), \tag{C.1}$$

where $\sigma_i^{(k)}$ and $\sigma_j^{(k)}$ are the Euclidean distance of data points $\boldsymbol{x}_i$ and $\boldsymbol{x}_j$ to their corresponding $k$-th nearest neighbors, respectively. In our experiments, we tune the parameter $k$ on each dataset for best clustering accuracy.

For Kernel SSC (KSSC) [56], AE+LSR, and SEDSC, we report the results obtained from [58, 68].

For subspace clustering methods including SSC [11], LSR [24], LRSC [107], EnSC [14], SSCOMP [13], $S^3$COMP [40], as well as manifold clustering methods including LLMC [94], SMCE [95], we produce the experimental results using the codes provided by the respective authors and tune the parameters for the best clustering accuracy.

For autoencoder + subspace clustering methods and deep subspace clustering methods, we cite the results reported in corresponding papers.

## C.3. Implementation Details for DELVE

For our DELVE, the number of layers $L$ and $\{(r^{(\ell)}(\cdot), \gamma^{(\ell)}, \kappa^{(\ell)}(\cdot, \cdot))\}_{\ell=1}^L$ in the self-expressive layers are subject to design choices. For simplicity, we use $r^{(\ell)}(\cdot) \doteq r(\cdot)$, $\gamma^{(\ell)} \doteq \gamma$, and $\kappa^{(\ell)}(\cdot, \cdot) \doteq \kappa(\cdot, \cdot)$, with $r(\cdot)$ chosen to be the elastic net regularizer in (4) and $\kappa(\cdot, \cdot)$ chosen to be (5), unless specified otherwise. Then, the parameters $L, \gamma$, as well as $\lambda$ in (4) and $\eta$ in (5) are hyper-parameters tuned for each dataset. To train our DELVE, we use the LARS solver [84] in the SPAMS package available at http://thoth.inrialpes.fr/people/mairal/spams/ for COIL100 and MNIST, and an Alternating Direction Method of Multiplier (ADMM) algorithm modified from the implementation provided in [83] for EYaleB.

In Table 6, we provide the parameters used in DELVE and related methods for experimental results reported in Table 4 and Table 2. The parameters include the following.

- $\lambda$: The trade-off parameter in the elastic net regularizer (4);

- $\gamma$: The trade-off paraemter in the self-expressive loss (3);

- $\eta$: The trade-off parameter in the penalty term (5);

- $L$: the number of layers of DELVE;

- $k$: parameter used in SMCE.

| Methods | Extended Yale B | | | | | COIL100 | | | | | MNIST | | | | |
|---|---|---|---|---|---|---|---|---|---|---|---|---|---|---|---|
| | $k$ | $\lambda$ | $\gamma$ | $\eta$ | $L$ | $k$ | $\lambda$ | $\gamma$ | $\eta$ | $L$ | $k$ | $\lambda$ | $\gamma$ | $\eta$ | $L$ |
| SSC | - | 1 | 30 | 1 | - | - | 1 | 15 | 0 | - | - | 1 | 3 | 0 | - |
| **DELVE**-SSC | - | 1 | 30 | 1 | 4 | - | 1 | 10 | 0 | 3 | - | 1 | 3 | 0 | 2 |
| EnSC | - | 0.9 | 30 | 1 | - | - | 0.90 | 3 | 0 | - | - | 0.3 | 0.5 | 0 | - |
| **DELVE**-EnSC | - | 0.9 | 30 | 1 | 6 | - | 0.95 | 30 | 0 | 11 | - | 0.3 | 0.5 | 0 | 7 |
| SMCE | 400 | 1 | 30 | 0 | - | 7 | 1 | 10 | 0 | - | - | - | - | - | - |
| **DELVE**-SMCE | 400 | 1 | 30 | 0 | 2 | 20 | 1 | 10 | 0 | 5 | - | - | - | - | - |

Table 6: Parameters used in each algorithm on each dataset for results reported in Table 4 and Table 2. "-" means that the method does not have this parameter.

In Table 7, we provide the parameters used in DELVE and related methods for experimental results reported in Table 5.

| Methods | CIFAR-10 | | | | | FashionMNIST | | | | |
|---------|---|---|---|---|---|---|---|---|---|---|
| | $k$ | $\lambda$ | $\gamma$ | $\eta$ | $L$ | $k$ | $\lambda$ | $\gamma$ | $\eta$ | $L$ |
| EnSC [14] | - | 0.8 | 200 | 0 | - | - | 0.9 | 100 | 0 | - |
| **DELVE**-EnSC | - | 0.5 | 200 | 0 | 8 | - | 0.9 | 200 | 0 | 3 |

Table 7: Parameters used in each algorithm on each dataset for results reported in Table 5. "-" means that the method does not have this parameter.

## D. Broader Implications and Limitations

There are broader implications of constructing deep network architecture from the self-expressive models. Note that self-expressive models are extensively studied in the past decade, in terms of both theoretical guarantee for correctness and algorithms for handling practical data (see Section 2 for a review). Such results provide the arsenal for better understanding and further improving DELVE, in ways that are not viable for standard deep neural networks.

Meanwhile, as the first work on constructing a novel deep network architecture from self-expressive models, this paper focuses on delineating relevant background and promises of the approach and on showing proof-of-concept experimental results, but fall short of providing a comprehensive study of this new framework and addressing all theoretical / practical challenges. In particular, we face at least the following immediate challenges.

**Theoretical challenges.** When the underlying low-dimensional manifolds are linear or affine, there exist well-established theoretical conditions (such as those based on subspace angle and point distribution) on whether a self-expressive model can successfully recover the manifolds. With the underlying manifolds being potentially nonlinear, it is natural to ask whether theoretical conditions can be established for the correctness of DELVE as well.

In particular, the theoretical guarantee for correctness of DELVE may be established if we have affirmative answers to the following two questions:

- Does the self-expressive model finds manifold-preserving representations (i.e., each point is represented only by others from its own manifold) for data in nonlinear manifold?
- Suppose that the representation matrix is manifold-preserving, does the self-expressive map "linearizes" the manifold?

For the first question, existing study [81] shows that even if the data points do not perfectly lie in linear subspaces but are corrupted by noise, self-expressive model still finds subspace-preserving solutions. Such a result may be generalized to the case where data lies in nonlinear manifolds, which we leave as a future study. For the second question, this paper provides intuitive argument and empirical evidence, but does not have theoretical justification yet.

**Practical challenges.** Some notable challenges with DELVE for practical applications are as follows.

1. **Inference.** While DELVE can be used to generate a linearized embedding for a given (training) dataset, it does not provide an explicit embedding function after training. Hence, it becomes nontrivial to apply a trained DELVE for generating embeddings for a set of new (test) data points. A naive strategy (as in many classical manifold learning techniques) is to add the new data points to the training set and retrain a DELVE model, which will produce an embedding for all data points. However, such a strategy has a high computational cost. Another idea is to parameterize each layer of DELVE by a self-expressive neural network [41] which, once trained, can be used to provide an embedding for any new (test) data. We leave a detailed study of this second idea to future work.

2. **Extension to (semi-)supervised learning.** DELVE is designed for unsupervised learning but in many practical cases the data may contain human annotations. When such annotation provides information on which points lie in the same manifold and which points do not, then it may be leveraged to produce self-expressive coefficients with better subspace-preserving properties (e.g., [108]). More broadly, annotation may be used to design supervised learning losses imposed on the output of DELVE to provide additional training signals for DELVE.

3. **Handling complex images.** Natural image is a particular case where the data can be modeled by a union of low-dimensional manifolds, hence DELVE can be used for performing image clustering tasks as shown in our experiments. However, images also contain additional patterns other than the low-dimensionality, and such additional patterns are usually pivotal for image related learning tasks particularly for images of high complexity and large size. Hence, effectively handling such image data with DELVE may require additional engineering designs, such as pixel or patch-level processing, invariance, data augmentation, and so on.

