# OpenReview forum: "Deep Self-expressive Learning"
_CPAL.cc/2024/Conference — CPAL 2024 (Proceedings Track) Oral_

### Official Review · Reviewer_gKan · 2023-10-05
**Review of Submission 42: Self-expressive Learning**

**Rating:** 8
**Confidence:** 3

**Review:**

## Overview:

In this paper, the authors extend self-expressive learning, which can be used to identify separate *linear*, low-dimensional manifolds that data lies in the union of, to the setting where the manifolds are non-linear. This captures a much wider class of data-sets, and through numerical experiments the authors show how their deep self-expressive approach can cluster with state-of-the-art accuracy on real world data like faces (EYaleB) and handwritten symbols (MNIST). Additionally, they show how their approach can handle corruption and can be further modified for other possible settings.

## Strengths:

1. This paper presents a novel approach to self-expressive learning that yields strong numerical results.
2. This paper is well aligned to the overarching goals of CPAL.
3. This paper is well written and easy to follow. I was unfamiliar with self-expressive learning, but the authors enabled me to easily follow along.
4. The illustrations of the approach (Figs. 1 and 2) were very helpful in understanding the method and made it easy to follow along.
5. The authors discuss several ways in which to use in their approach in other settings, such as by modifying the regularizer and penalty term.  I believe this will make it easy for others to implement, in an *intelligent* manner.
6. The authors present a comparison of their approach to self-attention and graph convolution networks, which I found to be interesting, informative, and topical.

## Weaknesses:

I found no major weaknesses of the paper.

However, I do think there are a few points that, if addressed. would make this paper stronger:

1. In Fig. 2, DELVE is compared to an autoencoder. While the representations learned by DELVE look much nicer (and support the authors' claims), the comparison to the autoencoder is complicated by the fact that only 4 layers of the autoencoder are shown, as compared to 40 DELVE layers. I assume the reason there aren't more layers of the autoencoder is because it is difficult to train one with 40 layers? Commenting on this rationale, as well as showing perhaps the DELVE representations after 2 or 4 layers (so as to at least match the autoencoder depth once in the figure), would be helpful in making a more clear comparison.
2. It would be helpful for the reader to know more about the data sets used in Sec. 4 (what is COIL100?), as well as the other methods that DELVE is compared to (what is NMCE, and is it known that ``The better performance of NMCE relies critically on its use of data augmentation'' [lines 241-242]? If so, is there a citation that could be referenced?). It would additionally be helpful to reference, in the main text, that there are Tables and extra details presented in the Appendices, so that the reader can know that more details of the experiments are reported elsewhere.
3. The paper ends rather abruptly, as there is no conclusion. I think it is reasonable to move some of Table 2 to the supplement and then have a paragraph or two re-iterating why this work is important and what future directions the authors foresee.

## Questions:

1. The choice of penalty term (Eq. 5) appears to be motivated by the elastic net regularizer (Eq. 4), but is never explicitly stated. Is that the correct motivation, or is there a separate reason for choosing that form?

## Summary:

This is a great paper that introduces a novel idea, that fits well within the CPAL CfP and yields strong experimental results. I believe would be a strong addition to the first annual CPAL.

---

### Official Review · Reviewer_LuGX · 2023-10-08

**Rating:** 7
**Confidence:** 2

**Review:**

**Summary**:
The paper proposes a self-expressive model for clustering data drawn from a union of nonlinear manifolds, extending methods working in the linear regime.

**Pros**:
- The paper is, in general, easy to follow and clearly written.
- The proposed method is a natural extension of a similar idea introduced for linear manifolds, and, within the provided in the paper empirical evaluation, is effective.


**Cons**:
- The method seems to be grounded in intuition from linear case. In consequence, I am not sure whether it is theoretically guaranteed to work
- The provided results could be complemented by a comment on the stability of the methods, as well as standard deviations.
- The text ends abruptly

(See the “Details” for more comments/reasoning behind the above pros and cons)

**Details**:

The paper is of good quality and clarity. I also consider the idea of extending the linear approach of self-expressive models to a nonlinear case by progressively linearizing the representations interesting. However, apart from the provided intuition given by the authors on page 2 I do not understand (or see) whether there any (even simplified) theoretical guarantees guiding this approach (not that I necessarily need to see ones, but would appreciate a comment on their existence in the text). I appreciate the provided evaluation with other approaches. However,I would like to have the blank spaces in Table 2 clarified (i.e. why some methods were not evaluated on some data). Additionally, I would also like to see a comment on the stability of the methods, preferably by adding information about deviation from the results reported in the Tables in the text. Finally, In general, I believe the paper is easy to follow and clearly written, however, it ends abruptly, without any remarks on the closing conclusions. I would like to see those issues addressed. Beyond the points above I do not see any major flaws in the paper.

---

### Official Review · Reviewer_yobb · 2023-10-15
**Progressive linearization and clustering of data drawn from a union of manifolds**

**Rating:** 7
**Confidence:** 2

**Review:**

The goal of this work is to obtain linearized representations and cluster data that is drawn from a union of low dimensional manifolds. Existing methods either have difficulties with less/noisy samples or use deep models that lack interpretability.

To achieve this task using interpretable deep models, the authors propose an approach called DELVE that takes inspiration from linear subspace clustering methods. The key idea used by DELVE is that a data point on a non-linear manifold can be approximated using a linear combination of points in its local neighborhood on that same manifold. This is used to build a sequence of ‘self-expressive layers’ that progressively linearize the input manifold. The parameters of each layer are sequentially optimized in an unsupervised manner by using a ‘self-expressive loss’.

**Strengths:**

1. DELVE achieves good clustering performance, comparable to existing deep models. The model can also scale well to large datasets since it does not necessarily require backpropagation through all the layers. Additionally, from Fig. 2, the obtained representations visually appear meaningful.

2. I like the white-box characteristic of DELVE. The role played by each layer is interpretable and one can play with different hyper-parameters and observe performance changes from the model that are mathematically explainable.

3. The paper is generally well written.

I have a few questions/concerns regarding DELVE that I would like the authors to clarify.

1. A key premise for DELVE to succeed is the ability to linearly approximate a data point using other points in its local neighborhood on the manifold. How would DELVE perform in comparison to an autoencoder if the ambient dimension/image size increases (or available data points from the manifold are farther apart from each other)? How do the obtained representations in such case look like? Are they still meaningful?

One of the limitations of conventional approaches like locally linear embeddings is their difficulty to deal with less samples. I wonder if DELVE would have a similar performance decline (if any) or its model depth could somehow help with that.

2. Once an autoencoder is trained, it can perform a quick forward pass to yield representations for a new data point. On the other hand, it would be quite inefficient if DELVE had to run the entire optimization program every time one needed representation for a new data point. How quickly can DELVE do this in comparison to a conventional autoencoder? Is there an efficient method that DELVE can utilize?

---

### Meta-Review · Area_Chair_E8wP · 2023-11-13

**Recommendation:** Accept (Poster)
**Confidence:** 4

**Metareview:**

This paper proposes a self-expressive model to cluster data drawn from a union of nonlinear manifold. It is a novel extension of the self-expressive model from the linear regime to the nonlinear regime. All reviewers found the proposed approach interesting and novel, and the proposed DELVE was supported by implementation details and experimental validation. I agree with the reviewers that this paper is of high quality, and recommend an acceptance.

---

### Decision · Program_Chairs · 2023-11-19

**Decision:**

Accept (Oral)

**Comment:**

The paper introduces DELVE, a self-expressive model for clustering data from nonlinear manifolds. Reviewers generally find the paper well-written and the idea of extending the linear self-expressive model to the nonlinear regime interesting. The paper receives praise for its clear presentation and empirical evaluation. However, some reviewers express concerns about the lack of theoretical guarantees and suggest including stability analysis and standard deviations in the results. Overall, both the reviewers and AC recommend acceptance, recognizing the paper's novelty and high quality.

The action PC chair for this paper is Atlas Wang, who made the decision after carefully reading the paper as well as the comments by all reviewers and AC. The decision is agreed by all PC chairs.